# Biological Scaffolds for Abdominal Wall Repair: Future in Clinical Application?

**DOI:** 10.3390/ma12152375

**Published:** 2019-07-25

**Authors:** Alessandra Costa, Sergio Adamo, Francesco Gossetti, Linda D’Amore, Francesca Ceci, Paolo Negro, Paolo Bruzzone

**Affiliations:** 1Sezione di Istologia ed Embriologia Medica, Dipartimento SAIMLAL, Sapienza Università di Roma, Via A. Scarpa 16, 00161 Rome, Italy; 2Dipartimento Assistenziale Integrato Cardio Toraco—Vascolare, Chirurgia e Trapianti d’Organo, Azienda Ospedaliera Universitaria Policlinico Umberto I. Dipartimento Universitario Chirurgia Generale e Specialistica “Paride Stefanini”, Sapienza Università di Roma, Viale del Policlinico 155, 00161 Rome, Italy

**Keywords:** hernia repair, abdominal wall, clinical application, biological biomaterials, ECM (extracellular matrix)

## Abstract

Millions of abdominal wall repair procedures are performed each year for primary and incisional hernias both in the European Union and in the United States with extremely high costs. Synthetic meshes approved for augmenting abdominal wall repair provide adequate mechanical support but have significant drawbacks (seroma formation, adhesion to viscera, stiffness of abdominal wall, and infection). Biologic scaffolds (i.e., derived from naturally occurring materials) represent an alternative to synthetic surgical meshes and are less sensitive to infection. Among biologic scaffolds, extracellular matrix scaffolds promote stem/progenitor cell recruitment in models of tissue remodeling and, in the specific application of abdominal wall repair, have enough mechanical strength to support the repair. However, many concerns remain about the use of these scaffolds in the clinic due to their higher cost of production compared with synthetic meshes, despite having the same recurrence rate. The present review aims to highlight the pros and cons of using biologic scaffolds as surgical devices for abdominal wall repair and present possible improvements to widen their use in clinical practice.

## 1. Introduction

Surgery for hernia repair represents the most common type of abdominal wall procedure performed today. Depending on the anatomical site, abdominal wall hernias can be classified into the groin (femoral or inguinal) or ventral hernias (umbilical, epigastric, Spigelian, lumbar, and incisional) [1]. Over one million repair procedures of both hernia types are estimated to have occurred in 2003 in the United States [2], with 400,000 procedures in Europe [3]. While the number of non-incisional hernias is constant, the prevalence of incisional hernia repairs is increasing yearly [4]. Health care costs in the US for both ventral and incisional hernia repairs are estimated at more than $10 billion/year, which is a cost that will increase over time because of increasing patient age and a consequential higher frequency of comorbidities and adverse events [5,6].

Complex Abdominal Wall Reconstruction (CAWR) is a demanding procedure, which requires appropriate surgeon training, correct timing of the procedure, accurate placement, and the correct choice of prosthesis.

Synthetic meshes, which are considered the gold standard for abdominal wall repair, represent a minor part of the cost. The most advantageous feature of synthetic meshes is their strength. Moreover, some meshes are designed to have a non-adherent side facing the viscera [4]. However, their major disadvantages include inflammation with consequent stiffness and abdominal pain, a high infection rate, adhesion to the viscera, and enterocutaneous fistulae [4,7]. Pros and cons of synthetic versus biologic meshes in hernia repair are summarized in Figure 1. Furthermore, even though the introduction of synthetic meshes in hernia repair drastically reduced the recurrence rate, it is still estimated at around 24% after three years [8]. In the last decade, meshes of biological origin have been studied as possible alternatives to synthetic materials. Biological meshes have shown many advantages, including a milder immune response, a decreased incidence of fistulae, and reduced fibrosis. However, there are some concerns with the use of biological materials such as their supposedly inferior mechanical strength and higher cost of production. In this paper, we highlight the pros and cons of the use of biological meshes in abdominal wall repair, as determined in both pre-clinical and clinical studies. This review does not consider synthetic meshes in detail, as they are extensively described elsewhere [9,10,11]. A table of synthetic and biological meshes, namely biomaterials derived from the extracellular matrix (ECM), is available below (Table 1).

Information sources for Table 1 are biomaterial datasheets furnished by the manufacturers and available at manufacturers’ websites (see Table 3).

## 2. Biological Scaffolds for Abdominal Wall Repair

### 2.1. Composite Meshes: Synthetic/Natural Biomaterials

In order to overcome the issues related to the application of synthetic meshes (i.e., polyester and polypropylene meshes), second-generation composite surgical meshes for hernia repair were developed [12]. Composite meshes have a synthetic layer facing the dermis, which provides mechanical strength and promotes collagen accumulation and ingrowth, and an internal layer coated with a naturally degradable biomaterial, facing the peritoneum, whose primary role is to avoid adhesion to the viscera [4] (Figure 2). The natural layer of such meshes is usually degraded within one to eight months [13].

The Parietex™ Composite (Covidien Sofradim Production), which is a polyester material coated with an absorbable hydrophilic collagen film, showed superior integration into the abdominal wall of a rabbit model when compared to a synthetic mesh [14]. A retrospective study involving patients subject to laparoscopic incisional and ventral hernia repairs highlighted that the use of Parietex™ significantly reduced adhesion to viscera and consequent bowel obstruction [15]. A polyester mesh coated with an absorbable layer of oxidized collagen and chitosan, facing the peritoneum, applied in a full-thickness abdominal wall repair in rabbits, led to the lowest incidence of seroma formation when compared to both synthetic mesh and other composite meshes. This property could be related to the degradation of the collagen film. At 90 days after implantation, the composite mesh was well integrated with the abdominal wall, which shows the de novo formation of peritoneal tissue with vascularization and minimal inflammation [16].

Clinical studies confirmed a reduction in adhesion when the composite mesh Parietex™ (77%) was used for hernia repair compared with uncoated mesh (18%) [17]. Non-comparative clinical studies evaluating the adhesion to viscera at the time of reoperation after hernia repair highlighted a rather low percentage of adhesion for double-faced synthetic meshes, such as DUALMESH^®^ Biomaterial by Gore (9%) [18] and Parietex™ (11%) [19].

A study by Schreinemacher et al. [20] performed in rats with a 90-day follow-up period suggested that the reduced adhesion to viscera, obtained by the application of composite synthetic/natural meshes, is independent of a reduction in the inflammatory response. All composite meshes used in this study—Parietene™ Composite (polypropylene/collagen), Parietex™, C-Qur Edge (polypropylene/omega-3 fatty acids), and Sepramesh™ (polypropylene/cellulose)—showed good prevention of adhesion to the viscera, despite having different inflammatory responses. The C-Qur Edge showed the lowest inflammatory response, with a moderate presence of macrophages and foreign body giant cells (FBGCs) and an absence of granulocytes at 90 days. The highest presence of macrophages and FBGCs was shown by Parietex™, followed by Parietene™, even though the adhesion was not affected in either of these meshes. The authors hypothesized that the surface of the natural layer is smooth, which prevents tissue anchoring, regardless of the inflammation level. Similarly, although C-Qur Edge implants had the lowest inflammatory response, this did not lead to reduced adhesion when compared with the other composite meshes [20]. The observation that the severity of the inflammatory response does not correlate with negative performance of composite meshes (i.e., increased adhesion), which was also investigated by a vast body of literature [21,22,23,24]. This supports the hypothesis that, in several tissues and organs, during the repair process, the activation of the inflammatory response is a crucial event. Furthermore, after the first pro-inflammatory phase due to injury, the shift toward an anti-inflammatory environment promotes constructive tissue remodeling (i.e., the differentiation of progenitor/stem cells, the formation of a mature vasculature, and the secretion of a new extracellular matrix), while a persisting pro-inflammatory scenario leads to tissue repair failure. A table of composite meshes available on the market is available below (Table 2).

Information sources for Table 2 are biomaterial datasheets furnished by the manufacturers and available at manufacturers’ websites (see Table 3).

### 2.2. ECM Biomaterials

ECM biomaterials are obtained by decellularization, i.e., the removal of the cellular component from tissues and organs. The decellularization process can be achieved by multiple methods based on the use of physicochemical agents and enzyme action [25]. A good decellularization process is essential in order to achieve cell removal while maintaining the native ECM in a state that is as intact as possible, to avoid rejection and to promote constructive tissue remodelling [26]. ECM has been demonstrated to play a fundamental, active role during tissue remodelling after injury, which promotes tissue healing, and an anti-inflammatory environment in various tissues [27,28,29,30,31,32]. Together with growth factors (GFs), ECM degradation products released during remodelling are thought to be responsible for ECM bioactivity [33,34,35,36]. In addition to the mentioned advantages offered by ECM biomaterials, the antibacterial action of such biomaterials has been demonstrated [37,38,39,40].

#### 2.2.1. Antibacterial Action

The ability of ECM biomaterials to resist infections appears to be particularly helpful in hernia repair conducted in a contaminated field. However, confounding data have emerged on the antimicrobial properties of ECM biomaterials used in hernia repair from pre-clinical and clinical studies. A pre-clinical study analyzing the microbial clearance against *S. aureus* and *E. coli* of a biologic mesh (Strattice™, ECM from porcine pericardium by Allergan), vs. a macro-porous synthetic one (Progrip™) implanted subcutaneously demonstrated better performance of Progrip™ against *E. coli* infection but not against *S. aureus* [41]. Another pre-clinical study by Harth et al. [39] demonstrated remarkably better clearance against *S. aureus* for all biologic meshes used for hernia repair (Permacol™, Surgisis^®^, Xenmatrix™, and Strattice™) compared to Parietex™ (synthetic mesh). A recent retrospective study conducted by Koscielny et al. [42] compared two groups of 28 patients each who underwent hernia repair with an ECM biomaterial derived from small intestinal submucosa (SIS) (Surgisis/Biodesign^®^ by Cook Medical) or with different polypropylene meshes (12 Ultrapro^®^ and 12 Vypro, both by Ethicon, J&J Medical). Patients were pair-matched for eight criteria (gender, age, comorbidity profile, body mass index, type of hernia, mesh implantation technique, CDC (Centers for Disease Control and Prevention) wound class, and source of potential contamination/primary surgery performed) for a minimum follow-up period of 24 months. Furthermore, an analysis of the hospitalization duration, hernia recurrence, and negative occurrence at the operation site was conducted. The hospitalization duration and hernia recurrence almost doubled for patients with SIS biomaterial, and the occurrence of hematoma and necrosis was also significantly higher than in patients with synthetic meshes. However, even though this study distinctly suggests worse outcomes with the use of SIS biomaterial, the study limitations, namely the small number of patients recruited in a single study center, must be considered.

Studies focused on the antimicrobial activity of mesh materials in hernia repair suggest that ECM biomaterial mesh performance is mostly dependent upon the type of bacterial infection rather than the origin of the ECM. In contrast, synthetic mesh performance is more dependent upon the type of mesh used. This observation is a crucial evaluation point for surgeons when choosing the best mesh material. Yet, a single change in an ECM biomaterial, such as cross-linking, can completely change its clinical outcome, as demonstrated by two clinical studies on PADM (porcine acellular dermal matrix). Majumder at al. [43] conducted a multicenter retrospective study involving 129 patients, in which 69 patients were treated with non-cross-linked PADM and 67 patients were treated with synthetic polypropylene meshes in a clean-contaminated or contaminated field, and a higher infection occurrence (31.9% vs. 12.3%) and more frequent recurrence (26.3% vs. 8.9%) were found for patients receiving PADM when compared with synthetic mesh over a follow-up period of 20 months. Conversely, a clinical trial by Gossetti et al. [44] with characteristics similar to the previous study conducted on 36 patients using PADM cross-linked with hexamethylene diisocyanate, showed a lower infection occurrence (19%) and a recurrence rate of 2.8% with a follow-up period longer than five years. The authors, thus, showed that a single change in ECM-biomaterial can influence clinical outcomes.

#### 2.2.2. Recurrence Rate and Mechanical Strength

The preclinical studies described so far highlighted a higher recurrence rate for ECM biomaterial meshes when compared to synthetic meshes [42,43]. A recent retrospective study of 229 patients revealed recurrence rates of 6.9%, 11.2%, and 22.0% for Fortiva^®^ (porcine dermal ECM by RTI Surgical, Inc.), Strattice™, and Alloderm^®^ (human dermal ECM by LifeCell), respectively, with a median follow-up time of 20.9 months, which suggests that ECM biomaterials are durable meshes for hernia repair [45]. Another clinical study of 223 patients who underwent open ventral hernia repair analyzed the clinical outcomes of three human dermal ECMs, Alloderm^®^, AlloMax™ (C.R. Bard), and FlexHD^®^ (Ethicon), and two porcine dermal ECMs, Strattice™ and Xenmatrix™ (by C.R. Bard). The recurrence rate after 18 months varied between the types of ECM biomaterial implanted: 35% for Alloderm, 34.5% for AlloMax, 37.1% for FlexHD, 14.7% for Strattice, and 59.1% for Xenmatrix. All ECM biomaterials showed low infection rates [46]. The observed high recurrence for ECM biomaterial meshes is commonly ascribed to the degradable nature of the ECM and the consequent mechanical weakness when compared with synthetic mesh. However, the degradation rate of the ECM during tissue remodeling is also influenced by the deposition of new ECMs by the cells that colonize the implanted ECM biomaterials. Thus, ECM degradation is also strictly dependent upon the ability of the ECM itself to recruit stem/progenitor cells, to modulate the inflammatory response, and to respond to mechanical stimuli. A recent preclinical study demonstrated that a non-cross-linked ECM (six-layer urinary bladder matrix), implanted to repair a full-thickness abdominal wall defect in rats, degraded by 50% in 26 days, and completely degraded in 90 days. Conversely, its mechanical properties, measured as peak stress (from 0.39 ± 0.05 MPa at day 21 up to 0.93 ± 0.08 MPa at day 180), and strain (between 5% to 10% at 7 days up to 20% to 25% at day 180), showed gradual increases after day 20, which may be associated with the deposition of a new matrix, since the implanted ECM continued to degrade. Importantly peak stress, modulus, and strain after day 7 post-implantation were always similar to the values measured for native abdominal wall skeletal muscle. The non-cross-linked urinary bladder matrix (UBM) also showed better tissue integration and remodeling [47]. The mechanisms described above may explain why different ECMs (varying in animal or tissue source) and/or different configurations (varying in the decellularization process or final customization, i.e., single/multi-layer) lead to different clinical outcomes.

Cross-linking is a common strategy used to reinforce the mechanical strength of ECM biomaterials, which avoids untimely degradation [48,49]. In a preclinical study comparing cholecyst-derived ECM cross-linked by carbodiimide with Surgisis^®^, Peri-Guard^®^ (cross-linked bovine pericardium), and Prolene^®^ at 28 and 56 days after implantation, all biomaterials tested prevented adhesion to viscera. Prolene^®^ caused the shrinkage of the operated site, while the area repaired with Surgisis^®^ was subject to stretching (from 5% at 28 days to 48% at 56 days) because of a quick degradation, whereas the cross-linking of the cholecyst-derived ECM delayed the degradation, which reduced stretching and increased the volume fraction of collagen [50]. De Castro Bràs et al. compared the integration of two non-cross-linked ECMs (Alloderm^®^ and Surgisis™) and two cross-linked ones (CollaMend™ and Permacol™) in a rat model of abdominal hernia repair. Surgisis™ stimulated a severe inflammatory response, while Alloderm^®^ showed moderate to severe inflammation, an increasing maximum load from three to six months, and functional integration with the host tissue. For the cross-linked ECMs, CollaMend™ showed very poor integration, while Permacol™ caused a low inflammatory response, which was resolved and led to the best integration with the host tissue [51]. However, the cross-linking of ECMs represents a double-edged sword, since cross-linking can improve the mechanical strength of the ECM, decreasing its degradation rate, but hindering degradation can also prevent the release of matricryptic peptides and GFs, which are the key activators of tissue remodeling [47,52]. Furthermore, it is important to highlight that the comparison of mechanical properties of different meshes for abdominal wall repair is not always easy, due to the wide range of tests that can be performed. Moreover, most of the studies reported uniaxial tests, while multi-axial tests should be preferred for a tissue that is subject to stress and loading from more than one direction. Deeken and Lake produced a detailed discussion of the mechanical properties of biomaterials used in abdominal wall repair [53].

#### 2.2.3. Costs and Benefits

The production and use of ECM biomaterials as meshes for hernia repair are associated with a higher cost compared to synthetic meshes. Thus, the question is whether the use of ECM biomaterials is associated with benefits in clinical practice. Few studies have attempted to answer this question. Determining the costs and benefits of the application of a selected biomaterial toward another can be challenging. Biomaterial prices vary widely among different countries and among different time periods. Moreover, biomaterials can change because of the optimization processes. However, the price of the biomaterial is just a small percentage of the total cost, which includes the implantation procedure, the hospitalization time, and several post-hospitalization controls. Evaluation of the benefits can also be challenging. A cost-efficacy analysis [54] could be optimal to make the best choice between two options for the same clinical scenario, but it would be difficult to apply to the choice between synthetic and biological meshes. In fact, these are used in different clinical situations, such as for clean low-grade hernias, and contaminated high-grade hernias, respectively. Thus, a cost-utility analysis is more appropriate, to take into account the patient’s quality-adjusted life years [54]. However, as reported in this review, there are few clinical studies with a follow up longer than five years and many studies are retrospective, so information about patient’s quality of life or recurrence after five years is missing.

Nissen et al. carried out the Hernia-Related Quality of Life Survey [55] on 175 patients who underwent ventral hernia repair at Stanford University Medical Center (1998–2017) by means of synthetic meshes, ECM biomaterials, or suture only. The main factors negatively affecting the quality of life of these patients were hernias greater than 50 cm^2^, obesity, tobacco use, previous abdominal surgery, hernia recurrence, and postoperative complications. The quality of life was lower for patients treated with ECM biomaterials than for those treated with synthetic meshes for grade 2 hernias (Modified ventral hernia working group grading system) [56], while there was no significant difference between the two types of mesh for grade 1 hernias [57]. Schneeberger et al. built up a cost-utility analysis model based on published studies, health care costs, utilization project data, and Americas Hernia Society Quality Collaborative data. This model showed that synthetic meshes are the best option in terms of cost-utility for a base case scenario with a five-year follow-up, but the use of biological meshes could be the best choice over the long term since the complication rate of synthetic meshes beyond five years increases up to the point that the higher initial cost is justified [58]. However, cost–utility analyses remain difficult since synthetic meshes are used in clean low-grade hernias, which leads to excellent clinical outcomes with a low cost, while biologic meshes are usually implanted in contaminated, higher-grade hernias or following hernia recurrence—cases that justify the high cost—leading to controversial outcomes that can be affected by the compromised preoperative scenario. A list of manufacturers’ websites of all cited meshes is available in Table 3.

### 2.3. Next-Generation Hybrid Biomaterials

According to the literature and current clinical practice, the future of CAWR strongly depends on the improvement of biomaterials. The different nature of meshes used for ventral hernia repair confers specific properties to biomaterials, which makes these more suitable for a specific issue or clinical situation. We have seen that composite meshes (synthetic/natural) successfully solve the problem of adhesion to the viscera. Synthetic meshes represent an excellent treatment choice for non-contaminated grade I hernias and are less expensive than ECM biomaterials. The latter materials, on the other hand, are the best choice for the repair of grade II hernias in contaminated fields, are particularly resistant to *S. Aureus* infections, and promote tissue remodeling, but they can be subject to untimely degradation and are expensive. Thus, the perfect mesh for hernia repair still does not exist. An innovative path worth exploring is the optimization of the mentioned biomaterials to improve their properties. An example we already cited is the cross-linking of ECM biomaterials, which reduces the degradation rate once implanted in vivo but can also hinder the release of the molecules responsible for the constructive remodeling promoted by ECM degradation. In a study of the improvement of synthetic meshes, Reinbold et al. loaded polypropylene mesh with degradable poly (lactide-co-glycolide acid) microspheres carrying rifampicin, which is an antibacterial agent. The microspheres released rifampicin over 60 days and allowed for complete protection against *S. Aureus* in vivo [59]. This strategy could lead the way to develop unlimited loading combinations with not only antibiotics but also growth factors and cytokines in order to guide stem cell recruitment, tissue remodeling, and the inflammatory response. Biomaterial cell loading is another interesting strategy to improve biomaterial performance, e.g., mesenchymal stem cells loading could represent an alternative anti-microbial agent to drug loading [60] or a scaffold loaded with Adipose-Derived Stromal Vascular Fraction Cells in breast cancer reconstruction, which led to increased vascularization and the consequently prolonged survival of the graft [61]. When a biomaterial is loaded with cells, it is important to monitor both cell and biomaterial fate through non-invasive techniques, if possible [62].

A hybrid synthetic/ECM mesh, OviTex by TELABio^®^, is being tested in a multicenter, post-market, single-arm, prospective study (the BRAVO* study) to evaluate its performance in ventral hernia repair. OviTex is a multilayered ECM derived from ovine rumen together with interwoven polypropylene, which provides long-term strength and an improved load-sharing capability. Initial data from the BRAVO study (24 patients with a 90-day follow-up period) show no need for the removal of the device, and no cases of recurrence or device failure (data available at http://www.telabio.com). OviTex has also been applied in incisional herniorrhaphy in a contaminated (CDC Class IV) operative field. A 56-year-old male presented with a draining purulent periumbilical wound. The patient previously had three hernia recurrences, with the last repaired with a synthetic mesh that became infected. The patient developed a fistula-producing intestinal content and a necrotizing soft tissue infection. After a four-month follow-up period, the OviTex was demonstrated to support the integrity of the repair, wound granulation, and revascularization with consequent tissue repair [63]. OviTex was also tested in a case series for inguinal hernia repair, where it was used for inguinal hernia repair in 31 patients with a follow-up period of between three and 18 months. The results showed no recurrences, no complications (seromas or hematomas), no surgical site infections, and no incidence of Chronic Postoperative Inguinal Pain [64]. An important advantage of this strategy that the authors failed to mention is that the coupling of an ECM biomaterial with a synthetic support can avoid ECM cross-linking, which can be detrimental for tissue remodeling, as previously discussed. Moreover, the coupling of synthetic material with a variety of ECMs from different tissue sources can generate new specific tools suitable for different clinical situations (i.e., contaminated/clean hernias, different severities, and different anatomical sites).

A step forward in future research aiming to find the best mesh in hernia repair could reside in the field of smart biomaterials. For a detailed discussion of the principles underlying the creation of smart biomaterials, the authors suggest the review by Khan and Tanaka [65]. A smart biomaterial can change its configuration in response to one or more stimuli, e.g., temperature, pH, light, and electro/magnetic fields. The configuration change can lead to molecule release, cell recruitment, or a switch in the biomaterial configuration (e.g., transition from aqueous solution to hydrogel). Smart biomaterials have been created for cell seeding and co-culture, osteoblast adhesion and proliferation, drug, protein, DNA, and cell delivery, and cartilage and neural tissue engineering [65]. Particularly helpful in the abdominal wall repair application could be a controlled drug delivery and the ability to recruit progenitor cells from the surrounding tissue, but, to the best of our knowledge, no smart biomaterial has yet been studied for hernia repair.

## 3. Future in Clinical Application

The increase in the number of abdominal wall repair procedures represents an increasing cost for health care systems. Determination of the best mesh material is essential, in order to avoid hernia recurrence and negative occurrences (i.e., seroma and adhesion to viscera), reduce health care costs, and ameliorate patients’ quality of life. This review highlighted the advantages and disadvantages of each category of mesh currently available for abdominal wall repair. There is consensus that synthetic meshes should be applied in clean, low-grade hernias, and biological meshes should be used in contaminated, high-grade hernias. However, this choice and the consequent clinical outcomes are mainly dependent upon the experience and knowledge of the surgeon. Clinical studies involving a more significant number of patients for an adequate follow-up period and with the possibility of comparison between different mesh categories will help elucidate scenarios in which synthetic or biological mesh represents the best choice.

A major problem in the comparative study of the literature about meshes for CAWR is the rapid evolution of such products and the frequent changes of their trade names and/or of the producing company, because of commercial reasons (merging, acquisitions, etc.).

Therefore, clinical and experimental data on Mesh-Related Visceral Complications (MRVCs) are scarce, including data on adhesions after the intraperitoneal or pre-peritoneal positioning of the mesh. According to the authors’ experience in rats, compared to simple meshes, porous meshes could be a better choice for retro-rectus pre-peritoneal prosthetic ventral hernia repair. However, non-porous meshes are preferred for intraperitoneal use [66]. MRVCs, which increase during the post-operative period, are observed after inguinal hernia repair [67,68] as well as after CAWR [69], and are more frequent and severe after intraperitoneal placement, even for composite meshes. Different surgical techniques have been developed to partially solve the problem, such as covering the intraperitoneal mesh with a peritoneal flap or with other tissue, but such techniques have not shown optimal results, and, therefore, this topic is still under discussion [70]. In order to avoid MRVCs, only one center [71] has reverted to the use of cadaveric human *fascia lata* segments, sutured together, for CAWR [72]. Probably the more promising clinical advances in the field of CAWR will be robotic surgery and the further improvement of materials aimed at reducing MRVCs.

New pathways that present countless possibilities to be tailored for a variety of clinical scenarios are the study of next-generation hybrid materials combining synthetic and biological properties and the development of smart meshes that can change their properties in response to a plethora of stimuli, which adapts to any clinical need. A schematic summary of this review is presented in Figure 3.

Images of the individual mesh products, prices and other commercial information, and references can be found at the listed websites.

## Figures and Tables

**Figure 1 materials-12-02375-f001:**
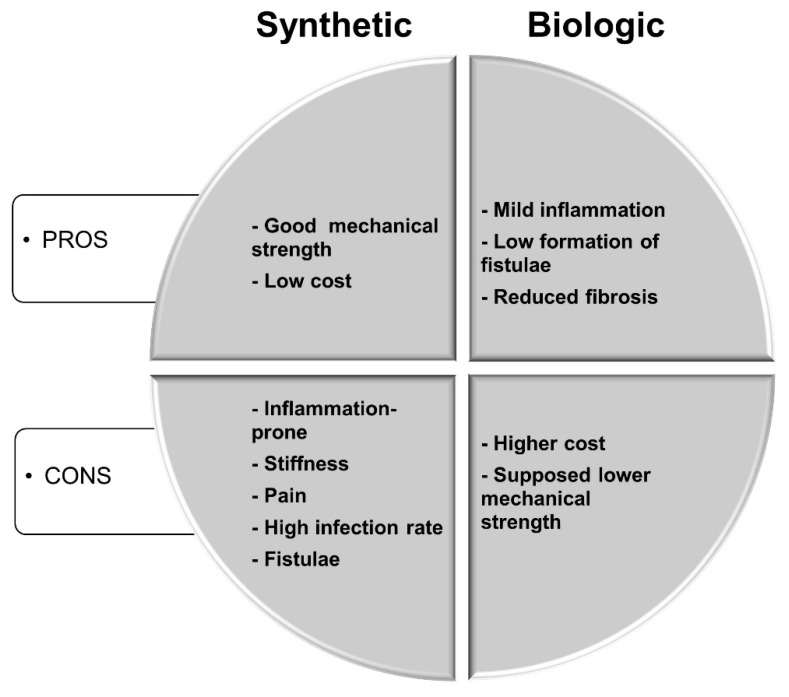
Schematic representation of pros and cons of synthetic versus biologic meshes in hernia repair.

**Figure 2 materials-12-02375-f002:**
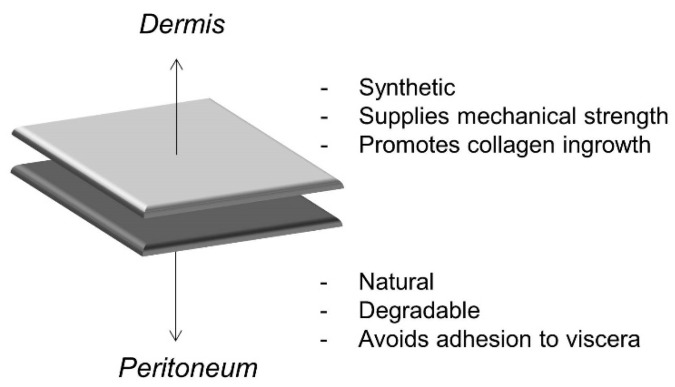
Schematic representation of the structure of composite meshes.

**Figure 3 materials-12-02375-f003:**
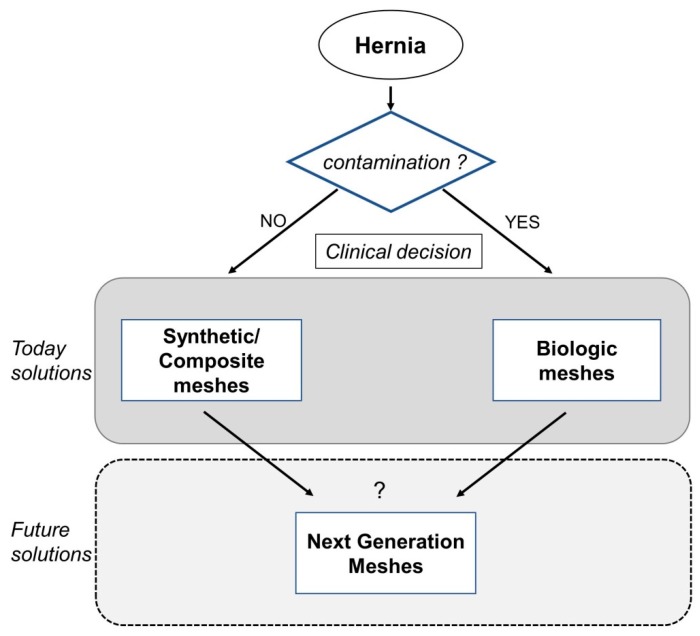
Chart summarizing the conclusions of this review.

**Table 1 materials-12-02375-t001:** Commercially available synthetic and ECM-biomaterial meshes for hernia repair.

**Synthetic Meshes**				
**Name**	**Manufacturer**	**Type of Material**	**Characteristics**	**Recommended Application**
Bard^®^ soft mesh	Bard Davol	Polypropylene	Monofilament	Repair of inguinal hernia
CK™ Parastomal hernia patch	Bard Davol	Polypropylene/ePTFE	Parietal side: Monofilament polypropylene for tissue ingrowth, Visceral side: ePTFE to minimize tissue attachment	Repair of parastomal hernias
CuraSoft™ patch	Bard Davol	PTFE mesh/ePTFE		Repair of hiatal and para-esophageal hernias
Dulex™	Bard Davol	ePTFE	Micro-porous side to minimize visceral attachment to the prosthesis and a macro-porous side to promote tissue in-growth	Hernia and soft tissue repair
Dualmesh Biomaterial^®^	Gore	ePTFF	Double face: Textured/soft, to avoid adhesion to viscera	Hernia and soft tissue repair, temporary bridging of fascial defects
PerFix™ plug	Bard Davol	Polypropylene	Monofilament	Repair of inguinal/groin hernias
Prolene^®^	Ethicon	Polypropylene		Small abdominal wall hernia repair
Ultrapro^®^	Ethicon	Monocryl/polypropylene filament	Monocryl is absorbable	Hernia repair
Ventralex™	Bard Davol	Polypropylene/ePTFE	Self-expanding: Eliminate the lateral dissection required for pre-peritoneal placement	Repair of umbilical hernias
Ventrio™	Bard Davol	Polypropylene/ePTFE	Parietal side: two layers of monofilament polypropylene Visceral side: Submicronic ePTFE, minimizing tissue attachment	Hernia and soft tissue repair
Visilex^®^ mesh	Bard Davol	Polypropylene	Monofilament Reinforced edges	Repair of inguinal hernias
3DMax™ mesh	Bard Davol	Polypropylene	3D	Repair of inguinal hernias
**ECM Biomaterial Meshes**				
**Name**	**Manufacturer**	**Species**	**Tissue**	**Recommended Application**
AlloDerm^®^	LifeCell	Human	Dermis	Repair/replacement of damaged/inadequate integumental tissues
AlloMax™	Bard Davol	Human	Dermis	Repair/replacement/reconstruction or augmentation of soft tissue. Horizontal/vertical soft tissue augmentation of thickness and length, such as post-mastectomy breast reconstruction.
CollaMend™	Bard Davol	Porcine	Dermis (cross-linked)	Hernia repair
Flex HD^®^	Ethicon	Human	Dermis	Complicated hernia repair
Fortiva^®^	RTI Surgical Inc.	Porcine	Dermis	Soft tissue reinforcement. Repair of damaged or ruptured soft tissue membranes. Repair of hernias and/or body wall defects.
Gentrix^®^ surgical matrix	ACell Inc.	Porcine	Urinary bladder	Repair of hernia, body wall, colon and rectal prolapse, esophagus
Peri-Guard^®^	Synovis Surgical Innovations	Bovine	Pericardium (cross-linked)	Repair of pericardial structures. Patch for intracardiac defects, great vessel, septal defect and annulus repair, and suture-line buttressing. Repair of soft tissue: defects of the abdominal and thoracic wall, gastric binding, muscle flap reinforcement, and hernias (diaphragmatic, femoral, incisional, inguinal, lumbar, para-colostomy, scrotal, and umbilical hernias).
Permacol™	Covidien	Porcine	Dermis (cross-linked)	Soft tissue repair. Hernia/abdominal wall repair.
Strattice^TM^	Life Cell Corporation (Allergan)	Porcine	Dermis	Soft tissue repair. Hernia/abdominal wall repair.
SurgiMend^®^	Integra LifeSciences	Bovine	Fetal Dermis	Soft tissue repair. Plastic and reconstructive surgery. Muscle flap reinforcement. Abdominal, inguinal, femoral, diaphragmatic, scrotal, umbilical, and incisional hernias.
Surgisis^®^/Biodesign^®^	Cook Medical	Porcine	Small Intestinal Submucosa (SIS)	Soft tissue repair. Dural repair. Hernia repair.
Tutopatch^®^	RTI Sugical	Bovine	Pericardium	Soft tissue repair. Plastic surgery. Repair of pericardial structures.
Veritas^®^	Synovis Surgical Innovations	Bovine	Pericardium	Reconstruction of the pelvic floor. Repair of rectal prolapse. Soft tissue repair: abdominal and thoracic wall repair, muscle flap reinforcement, and repair of hernia.
Xenmatrix™	Bard Davol	Porcine	Dermis (coated with a bioresorbable L-Tyrosine succinate polymer, which acts as a carrier for Rifampin and Minocycline)	Abdominal plastic and reconstructive surgery. Muscle flap reinforcement. Hernia repair.

**Table 2 materials-12-02375-t002:** Commercially available composite meshes for hernia repair.

Name	Manufacturer	Type of Material	Recommended Application
C-Qur meshes	Atrium	Polypropylene/Omega-3 fatty acids	Open and laparoscopic hernia repair
Parietene^TM^	Covidien	Polypropylene/Collagen film	Open and laparoscopic hernia repair
Parietex^TM^	Covidien	3D monofilament polyester (large pores)/Collagen film	Hernia repair
Proceed^®^	Ethicon	Polypropylene/Oxidized regenerated cellulose	Open and laparoscopic incisional hernia repair
Progrip™	Covidien	Polyester monofilament/absorbable micro grips of polylactic acid	Laparoscopic inguinal hernia repair
Sepramesh™	Bard Davol	Polypropylene/Cellulose	Hernia repair
Symbotex™	Covidien	Polyester/Collagen film	Abdominal wall repair
Vypro and Vypro II	Ethicon	Polypropylene/Polyglactin	Open and laparoscopic inguinal hernia repair

**Table 3 materials-12-02375-t003:** Web site listing of cited commercial meshes.

Commercial Meshes	Website
3DMaxTM mesh	https://www.crbard.com/CRBard/media/ProductAssets/DavolInc/PF10162/en-US/wazfaqlf5g21e0b9ps1vtsmfe8916fan.pdf
AlloDerm^®^	https://hcp.alloderm.com/
AlloMaxTM	https://www.crbard.com/davol/en-US/products/AlloMax-Surgical-Graft
Bard^®^ Soft Mesh	https://www.crbard.com/davol/en-US/products/Bard-Soft-Mesh
C-Qur meshes	http://www.atriummed.com/EN/Biosurgery/cqur-2.asp
CKTM Parastomal hernia patch	https://www.crbard.com/Davol/
CollaMendTM	https://www.crbard.com/CRBard/media/ProductAssets/DavolInc/PF10172/en-US/1e6bi57sa0gu9bpzp28xdr8ign3zo9yy.pdf
CuraSoftTM patch	https://akinglobal.com.tr/uploads/subdir-182-4/BARD_catalogue.pdf
Dualmesh Biomaterial^®^	https://www.goremedical.com/products/dualmesh
DulexTM	https://www.crbard.com/CRBard/media/ProductAssets/DavolInc/PF10149/en-US/gwe4a6bkum2biuyofjqh43f0asxud68x.pdf
Flex HD^®^	https://www.ethicon.com/na/epc/code/472225?lang = en-default
Fortiva^®^	http://www.rtix.com/en_us/products/product-implant/fortiva-porcine-dermis
Gentrix^®^	https://acell.com/surgical-matrix-products/
ParieteneTM	https://www.medtronic.com/covidien/en-us/products/hernia-repair/parietene-ds-composite-mesh.html
ParietexTM	https://www.medtronic.com/covidien/en-us/products/hernia-repair/parietex-composite-ventral-patch.html
PerFixTM plug	https://www.crbard.com/davol/en-US/products/PerFix-Plug
Peri-Guard^®^	http://www.baxterbiosurgery.com/us/resources/pdfs/periguard/PERI-GUARD_IFU.pdf
PermacolTM	https://www.medline.com/product/Permacol-Surgical-Implant-by-Covidien/Z05-PF40168
Proceed^®^	https://www.jnjmedicaldevices.com/en-US/search-hcp?search_api_fulltext = Proceed
ProgripTM	https://www.medtronic.com/covidien/en-us/products/hernia-repair/progrip-laparoscopic-self-fixating-mesh.html
Prolene^®^	https://www.ethicon.com/na/epc/search/keyword/PROLENE%C2%AE%20Polypropylene%20Mesh?lang = en-default
SeprameshTM	https://www.crbard.com/davol/en-US/products/Sepramesh-IP-Composite
StratticeTM	http://hcp.stratticetissuematrix.com/
SurgiMend^®^	https://www.surgimend.com/products/by-brand/surgimend/
Surgisis^®^/Biodesign	https://www.cookmedical.com/surgery/the-path-from-surgisis-to-biodesign/
SymbotexTM	https://www.medtronic.com/covidien/en-us/products/hernia-repair/symbotex-composite-mesh.html
Tutopatch^®^	http://www.rtix.com/en_us/products/product-implant/tutopatch-bovine-pericardium--tutomesh-fenestrated-bovine-pericardium
Ultrapro^®^	https://www.jnjmedicaldevices.com/en-US/product/ultrapro-hernia-system
VentralexTM	https://www.crbard.com/Davol/en-US/products/Ventralex-Hernia-Patch
VentrioTM	https://www.crbard.com/davol/en-US/products/Ventrio-Hernia-Patch
Veritas^®^	http://veritascollagenmatrix.com/
Visilex^®^ mesh	https://www.crbard.com/CRBard/media/ProductAssets/DavolInc/PF10172/en-US/1e6bi57sa0gu9bpzp28xdr8ign3zo9yy.pdf
Vypro / Vypro II	https://www.ethicon.com/na/epc?lang = en-default
XenmatrixTM	https://www.crbard.com/davol/en-US/products/XenMatrix-AB-Surgical-Graft

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
