# Peer review of "Biological Scaffolds for Abdominal Wall Repair: Future in Clinical Application?"

_materials, 2019, doi:10.3390/ma12152375_

Round 1

Reviewer 1 Report

This is a comprehensive review of the state of the art in the fabrication and application of biological scaffolds for abdominal wall repair.  The manuscript is thorough and presents a detailed review of previous studies dealing with ECM-based scaffolds and their applications in augmenting abdominal wall repair. I liked the fact that the authors summarized some of the published data in two Tables which collectively drive home some of the most key findings. Here are a few more comments to consider in order to further strengthen this well written manuscript before it can be published in Materials:

Consider adding references      to Tables 1 and 2 for each scaffold type.

In the description of some      of the results, it would be good if the authors added some images from some      of the various studies that they describe, so that the readers can      actually visualize some of the data. The way the data is presented now, it      is simply just text with numbers but as we all know, visual data is more      powerful.

Section 2.2.2, I recommend      that the authors include a table with some of the material property      numbers. Even though the title of this section contains the words “mechanical      strength” no numbers are given to indicate the actual material properties      of some of these scaffolds.  I am      sure that measurements on elasticity (i.e. elastic modulus; tensile      strength) at least have been reported in the literature and they should be      included.

Similarly, in section      2.2.3. Costs, the authors should provide some data as to how much do these      scaffolds cost in $$$? This would help the reader realize the actual cost      of just the scaffolds and not together with the cost of the surgical      procedure.

Author Response

·      Consider adding references to Tables 1 and 2 for each scaffold type.

The information given in Table 1 and Table 2 has been obtained from the datasheets of the biomaterials furnished by the manufactures and available at their own websites, where references for the individual products are also supplied. We could add a list of websites if the reviewer or the editor think it would be appropriate. Anyway, we have added a sentence to this effect below each Table.

·      In the description of some of the results, it would be good if the authors added some images from some of the various studies that they describe, so that the readers can actually visualize some of the data. The way the data is presented now, it is simply just text with numbers but as we all know, visual data is more powerful.

All these images are protected by Copyright – no Open Access. Requesting the permission to publish copyright-protected images would greatly delay the possible publication of our paper in this “Special Issue” of Materials. However, according to your suggestion, we have inserted into the text three schematic diagrams. 

·      Section 2.2.2, I recommend that the authors include a table with some of the material property numbers. Even though the title of this section contains the words “mechanical strength” no numbers are given to indicate the actual material properties of some of these scaffolds. I am sure that measurements on elasticity (i.e. elastic modulus; tensile strength) at least have been reported in the literature and they should be included.

This suggestion has been addressed discussed in Paragraph 2.2.2 (lines 14-19), and mechanical properties data has been added (see Paragraph 2.2.2, lines 39-40 and lines 42-44.

·      Similarly, in section 2.2.3. Costs, the authors should provide some data as to how much do these scaffolds cost in $$$? This would help the reader realize the actual cost of just the scaffolds and not together with the cost of the surgical procedure.

We have discussed this point in the appropriate section (2.2.3. Costs and Benefits).

Reviewer 2 Report

The authors discussed scaffolds for abdominal wall repair. They have compared synthetic and biological materials as well as their hybrids and provided an outlook for future including smart biomaterials. The authors paid a lot of attention to susceptibility to infections, but they missed to discuss anti-microbial properties of mesenchymal stem cells [1], which could be also used to improve the mechanical properties of scaffolds as it is in case of breast reconstruction [2]. The authors seemed also to miss a potential value of labeling and imaging of scaffold, which could potentially serve as an early indicator or biomarker of late performance [3]. At the beginning of the abstract the authors mentioned “hundreds of thousands”, which sounds weird and they could easily replace it with “millions” to what they actually refer in introduction.

1.            Mezey E and K Nemeth. (2015). Mesenchymal stem cells and infectious diseases: Smarter than drugs. Immunol Lett 168:208-14.

2.            Gentile P, D Casella, E Palma and C Calabrese. (2019). Engineered Fat Graft Enhanced with Adipose-Derived Stromal Vascular Fraction Cells for Regenerative Medicine: Clinical, Histological and Instrumental Evaluation in Breast Reconstruction. J Clin Med 8.

3.            Oliveira JM, L Carvalho, J Silva-Correia, S Vieira, M Majchrzak, B Lukomska, L Stanaszek, P Strymecka, I Malysz-Cymborska, D Golubczyk, L Kalkowski, RL Reis, M Janowski and P Walczak. (2018). Hydrogel-based scaffolds to support intrathecal stem cell transplantation as a gateway to the spinal cord: clinical needs, biomaterials, and imaging technologies. NPJ Regen Med 3:8.

Author Response

•          The authors discussed scaffolds for abdominal wall repair. They have compared synthetic and biological materials as well as their hybrids and provided an outlook for future including smart biomaterials. The authors paid a lot of attention to susceptibility to infections, but they missed to discuss anti-microbial properties of mesenchymal stem cells [1], which could be also used to improve the mechanical properties of scaffolds as it is in case of breast reconstruction [2]. The authors seemed also to miss a potential value of labeling and imaging of scaffold, which could potentially serve as an early indicator or biomarker of late performance [3]. 

1. Mezey E and K Nemeth. (2015). Mesenchymal stem cells and infectious diseases: Smarter than drugs. Immunol Lett 168:208-14.

2. Gentile P, D Casella, E Palma and C Calabrese. (2019). Engineered Fat Graft Enhanced with Adipose-Derived Stromal Vascular Fraction Cells for Regenerative Medicine: Clinical, Histological and Instrumental Evaluation in Breast Reconstruction. J Clin Med 8.

3. Oliveira JM, L Carvalho, J Silva-Correia, S Vieira, M Majchrzak, B Lukomska, L Stanaszek, P Strymecka, I Malysz-Cymborska, D Golubczyk, L Kalkowski, RL Reis, M Janowski and P Walczak. (2018). Hydrogel-based scaffolds to support intrathecal stem cell transplantation as a gateway to the spinal cord: clinical needs, biomaterials, and imaging technologies. NPJ Regen Med 3:8.

We thank the Referee for these useful suggestions, which we implemented.

•          At the beginning of the abstract the authors mentioned “hundreds of thousands”, which sounds weird and they could easily replace it with “millions” to what they actually refer in introduction.

Corrected in the abstract.

Reviewer 3 Report

This manuscript, containing not only introducing the current mesh materials for CAWR but also suggesting future perspectives, is well organized and comprehensively described. I recommend this work to be published in this journal.

In the Author contribution, there is missing of some authors in this manuscript.

I strongly recommend this manuscript has at least three or four Figures including a schematic diagram explaining a scaffold-based hernia repair process, which will help to non-expert readers a better understanding of this review paper. 

Author Response

•          In the Author contribution, there is missing of some authors in this manuscript.

Thanks for pointing out this omission, corrected.

•          I strongly recommend this manuscript has at least three or four Figures including a schematic diagram explaining a scaffold-based hernia repair process, which will help to non-expert readers a better understanding of this review paper.

All these images are protected by Copyright – no Open Access. Requesting the permission to publish copyright-protected images would greatly delay the possible publication of our paper in this “Special Issue” of Materials. However, according to your suggestion, we have inserted into the text three schematic diagrams. 

Round 2

Reviewer 1 Report

I thank the authors for their responses to my critique but unfortunately, it is disappointing that they did not incorporate some of the suggested recommendations so that their manuscript can be approved.  Specifically,

I am sure that manufacturers      may cite specific studies in their datasheet of websites that the authors      can certainly include in their tables. The sentence below each Table does      not suffice nor does it add anything valuable to the manuscript.

The authors claim that      including images is a problem because of copyright and requesting permission      to publish copyright-protected images would greatly delay possible      publication. As someone who has written several reviews, I know this is      not true.  Permission for copyrighted      material (including figures) is instantaneous and it is web based. I still      encourage the authors to do this as it will improve their own paper.

Although the authors state that they added text regarding costs in section 2.2.3., I still do not see a single number referring to cost. Can the authors provide in actual Dollars or Euros the actual cost of these products? I would think that this would not be such a difficult task.

Author Response

Dear Editor of Materials

Dear Reviewer

We thank the Reviewer again for his suggestions which we have carefully considered. We think that ours is a scientific paper, not a commercial price list: we deem the insertion of 34 pictures (the number of products we have listed) not feasible, and we do not see a criterion to select only some of them while leaving aside others.

We have provided a solution which substantially answers the Reviewer’s request, without the disadvantages outlined above: we have inserted a table listing the websites of all the products considered in the review. Visiting those websites allows readers to obtain images, commercial information (local representatives and prices), as well as references and other scientific information for each product.

We hope you will agree that, through intelligent use of the Referees’ comments, we have greatly improved the quality of our review in two revision rounds, and we do hope that our manuscript can now be considered publishable in this special collection of Materials.

Sincerely yours

Paolo Bruzzone MD